# Vaccination in Atherosclerosis

**DOI:** 10.3390/cells9122560

**Published:** 2020-11-30

**Authors:** Felix Sebastian Nettersheim, Lauren De Vore, Holger Winkels

**Affiliations:** Department III of Internal Medicine III, Heart Center, Faculty of Medicine and University Hospital of Cologne, 50937 Cologne, Germany; felix.nettersheim@uk-koeln.de (F.S.N.); lauren.de-vore@uk-koeln.de (L.D.V.)

**Keywords:** atherosclerosis, vaccination, immunization, low-density lipoprotein (LDL), apolipoprotein B (ApoB), antigen-specific, regulatory T cells (T_regs_), antibodies

## Abstract

Atherosclerosis is the major underlying pathology of cardiovascular diseases that together are the leading cause of death worldwide. The formation of atherosclerotic plaques is driven by chronic vascular inflammation. Although several risk factors have been identified and significant progress in disease prevention and treatment has been made, no therapeutic agents targeting inflammation are clinically available. Recent clinical trials established the potential of anti-inflammatory therapies as a treatment of atherosclerosis. However, adverse impacts on host defense have raised safety concerns about these therapies. Scientific evidence during the past 40 years implicated an adaptive immune response against plaque-associated autoantigens in atherogenesis. Preclinical data have underscored the protective potential of immunization against such targets precisely and without the impairment of host defense. In this review, we discuss the current vaccination strategies against atherosclerosis, supposed mechanisms of action, therapeutic potential, and the challenges that must be overcome in translating this idea into clinical practice.

## 1. Introduction

Atherosclerosis is a chronic inflammatory disease with an autoimmune component, which is characterized by formation of lipid-rich plaques in large and medium-sized arteries [1]. Progressive growth of plaques causes narrowing of the arteries, whereas plaque rupture or erosion with subsequent thrombus formation may lead to acute arterial occlusion. Together, these pathologies are the major causes of cardiovascular diseases (CVDs) including ischemic heart disease, cerebrovascular disease, and peripheral artery disease [2,3]. The development of effective therapies and prevention measures against infectious agents (such as antibiotics, vaccines, and modern hygiene concepts) replaced infectious diseases with CVDs as the leading cause of death worldwide [4]. The identification of major cardiovascular risk factors, such as hypercholesterolemia, arterial hypertension, cigarette smoking, and diabetes, as well as the subsequent development and widespread use of effective measures to prevent or treat CVDs (e.g., cholesterol-lowering and antihypertensive drugs, bypass surgery, and percutaneous vascular interventions), have led to a significant reduction in cardiovascular mortality in industrialized countries [3,4,5]. Nonetheless, CVDs still remain the leading cause of death worldwide [6]. The persistently high cardiovascular mortality rate might be explained by the fact that 30–50% of patients with CVDs are unexposed to traditionally associated risk factors [7,8], pointing to insufficient protection by the current available treatment options. Given that residual risk is mainly attributed to vascular inflammation [9], development of effective strategies to modulate the immune response involved in atherosclerosis is key to further reducing cardiovascular disease burden.

In 2017, the Canakinumab Antiinflammatory Thrombosis Outcome Study (CANTOS) trial demonstrated for the first time a beneficial effect of an anti-inflammatory treatment in CVD [10]. Canakinumab, a monoclonal antibody targeting the proinflammatory cytokine interleukin-1β (IL-1β), reduced major adverse cardiovascular events in patients with pre-existing coronary artery disease and an increased high-sensitivity C-reactive protein (hsCRP) level. Overall survival was not improved, and higher rates of fatal infections were observed [10], leading to this therapy not being approved for cardiovascular risk reduction. More recently, the Colchicine Cardiovascular Outcomes Trial (COLCOT) [11] and Low-Dose Colchicine (LoDoCo)2 [12] trials provided substantial evidence for a significant benefit of the anti-inflammatory drug colchicine in secondary prevention of CVD. Enrolling about 5000 patients each with recent myocardial infarction (COLCOT) or stable coronary artery disease (LoDoCo2), both randomized placebo-controlled trials demonstrated that low-dose colchicine therapy significantly reduced cardiovascular events without increased risk of fatal infections (in COLCOT, however, the risk of nonfatal pneumonia was significantly increased). Colchicine inhibits the NOD-, LRR-, and pyrin domain-containing protein 3 (NLRP3) inflammasome, a cytosolic multiprotein predominantly expressed in innate immune cells. NLRP3 is important for the release and formation of the pro-inflammatory cytokines IL-1β and IL-18, which have been associated with plaque formation and rupture [13,14]. Given its good safety profile, low price, and wide availability, colchicine emerges as a treatment option for secondary prevention of cardiovascular events. Nevertheless, a protective effect on cardiovascular diseases or overall mortality has yet to be demonstrated [11,12].

The adaptive immune system plays a critical role in atherogenesis [1]. In contrast to other medical conditions, such as cancer [15], immunomodulatory therapies for CVD are still not available in clinical practice. Preclinical evidence suggests that modulating the adaptive immune response involved in atherogenesis through vaccination―a strategy that has revolutionized the therapy of infectious diseases―might represent an attractive therapeutic approach [16,17,18,19,20,21,22].

## 2. The Role of Adaptive Immunity in Atherosclerosis

The initial steps of atherosclerotic plaque formation comprise endothelial dysfunction, accumulation of lipoproteins within the arterial wall, monocyte infiltration, oxidation of lipoproteins, and phagocytosis of these by macrophages, leading to foam-cell formation [1]. Shortly after monocytes were identified as the source to macrophage foam cells and a major component of atherosclerotic plaques [23], immunoglobulins [24,25,26], T cells [27], dendritic cells (DC) [28], and B cells [29] were detected in atherosclerotic lesions. Hyperresponsive natural killer (NK) cells have been detected in human atherosclerotic plaques [30], although they do not seem to contribute to experimental atherosclerosis [31,32]. Although macrophages represent the most common inflammatory cell type in atherosclerotic plaques, lymphocytes―in particular T cells―are present in significant numbers and have been identified as critical modulators of atherogenesis.

### 2.1. T Cells

T cells originate from bone marrow progenitor cells that migrate to the thymus, where they undergo maturation and selection. Sequential random recombination of the V, D, and J T-cell receptor (TCR)-α and TCR-β gene segments enables every T cell to obtain a unique TCR. The TCR is eventually expressed on the cell surface in association with the CD3 proteins in a complex which is essential for TCR signaling and thymocyte survival. Alongside TCR-α gene chain rearrangement, thymocytes upregulate expression of the TCR co-receptors CD4 and CD8 [33,34]. These so-called double-positive (DP) thymocytes then undergo a selection process whereby only DP cells with successfully rearranged TCR genes and surface expression of a TCR recognizing self-peptides in major histocompatibility complex (MHC) molecules with intermediate affinity survive. Subsequently, these cells downregulate one of the co-receptors and mature into single CD4^+^ (if selected against MHC class II) or CD8^+^ (if selected against MHC class I) T cells, which are finally released into periphery. Too strong or too weak binding results in apoptosis [35]. Once released, the mature yet antigen unexperienced (naïve) T cells circulate through secondary lymphoid organs to encounter their cognate antigen [36]. A functionally distinct subset of CD4^+^ T cells, natural regulatory T cells (nT_regs_), also referred to as thymic regulatory T cells (tT_regs_), also originates from the thymus. nT_regs_ comprise 5–10% of all peripheral CD4^+^ T cells and are already differentiated [37]. nT_regs_ are characterized by surface expression of the interleukin-2 receptor α chain (CD25) and the lineage-defining transcription factor (TF) forkhead box protein P3 (FoxP3). By utilizing several suppressive mechanisms including production of the anti-inflammatory cytokines interleukin-10 (IL-10) and transforming growth factor beta (TGF-β), they are able to control immune responses and are key protectors from autoimmunity [38].

In the periphery, CD4^+^ T cells recognize antigenic peptide sequences (so-called epitopes) bound to MHC class II molecules exclusively expressed on professional antigen-presenting cells (APCs), such as dendritic cells, macrophages, and B cells [39]. APCs internalize extracellular antigens and process these to short peptide sequences which are finally loaded on MHC-II molecules to be presented on the cell surface. CD8^+^ T cells recognize epitopes bound to MHC class I molecules that are expressed by all nucleated cells. These epitopes originate from intracellular proteins (e.g., viruses) [39]. In addition to TCR signals, CD4^+^ and CD8^+^ T cells require proper co-stimulatory and cytokine signals [40]. Successful activation of CD8^+^ T cells induces rapid proliferation and differentiation into cytotoxic T cells, which spread throughout the body to encounter and kill their targets (e.g., infected or tumor cells) [41]. Activated CD4^+^ T cells can differentiate into a variety of different CD4^+^ T cell effector subsets that may either promote or limit the immune response [42] depending on the integration of cytokine signaling, co-stimulatory signals, and the amplitude of the TCR signal which correlates to the binding strength to a given antigen [42]. CD4^+^ T cell subsets are characterized by expression of specific TFs, surface markers, and cytokines [43]. T helper 1 (T_H_1) cell differentiation is dependent on IL-12 and IL-18, expressing the TF T-box expressed in T cells (T-bet) and the pro-inflammatory cytokine interferon-γ (IFN-γ). TGF-β and IL-10 promote differentiation of anti-inflammatory T_regs_. These T_regs_ do not develop in the thymus but derive from naïve T cells in the periphery during an inflammatory response and are called inducible T_regs_ (iT_regs_) [44]. A specific subtype of iT_regs_ constitutes FoxP3-negative IL-10-producing type 1 regulatory (Tr1) cells that are characterized by surface expression of CD49b and lymphocyte activation gene 3 (LAG-3) [45]. Follicular helper T cells (T_FH_ cells), which express the TF B-cell lymphoma 6 (BCL-6), are critical for development and function of germinal centers and mediate selection of maturing B cells that produce antibodies with the highest affinity to a specific antigen [46,47]. T_H_1 cells are proatherogenic, whereas T_regs_ limit atherosclerosis [48].

Natural killer T (NKT) cells—well-known drivers of atherosclerosis—have a limited TCR repertoire and are activated by endogenous and exogenous lipid antigens presented in CD1d molecules [49,50].

However, the specific role of many CD4^+^ T cell subsets (e.g., T_H_2, T_H_9, or T_H_17 cells) in atherosclerosis is still unclear (for a thorough review of the role of CD4^+^ T cell subsets in atherosclerosis, the reader is referred to [48]).

In the late 1980s, Hansson and colleagues demonstrated that a substantial proportion of lesional T cells is activated [51]. They later successfully cloned oxidized low-density lipoprotein (oxLDL) responsive CD4^+^ T cells from human atherosclerotic plaques, which was the first evidence of a cellular autoimmune response in atherogenesis [52]. Initially, it was suspected that oxidation of LDL leads to formation of neoantigens, i.e., “altered-self” antigens that are recognized as “non-self” and, therefore, induce a T cell response [52]. More recently, the existence of CD4^+^ T cells reactive to epitopes derived from unmodified apolipoprotein B (ApoB), the core protein of LDL cholesterol and other lipoproteins, was demonstrated [53]. Some of these epitopes bound human and murine MHC-II [16,17,54,55]. Lastly, utilization of a T cell restimulation assay [55] and an innovative MHC-II-tetramer loaded with an ApoB-epitope [16] enabled direct detection of ApoB-reactive CD4^+^ T cells (ApoB^+^ T cells) in human blood from subjects with and without atherosclerosis. In healthy individuals, these ApoB^+^ T cells mainly comprised T_regs_. In contrast, patients with subclinical atherosclerosis harbored a substantial increase in T-bet and RAR-related orphan receptor gamma-T(RORγT)-expressing ApoB^+^ T cells, TFs typical of proatherogenic T_H_1 cells and T_H_17 cells [16]. Other autoantigens such as heat-shock proteins (HSPs) 60/65 [56,57,58] and pathogen-related antigens [59,60] have been detected in atherosclerotic plaques.

Taken together, there is profound evidence for the existence of T cells reactive to several atherosclerosis-related autoantigens, which predominantly exhibit an atheroprotective (T_reg_) phenotype in healthy individuals but frequently possess proinflammatory properties in patients with atherosclerosis. Suggesting that the latter may be causally involved in initiation and/or progression of the disease, the idea of a vaccination preventing the generation of these cells emerges as an attractive therapeutic approach.

### 2.2. B Cells

B cells develop in the bone marrow from hematopoietic stem cells in a multistage process that involves formation of a unique B-cell receptor (BCR), which is a membrane-bound immunoglobulin with signaling capacity [61]. Subsequently, they migrate to the spleen where further maturation into follicular (FO) B cells or marginal zone (MZ) B cells takes place [62]. Similar to T cells, maturing B cells undergo selection at several checkpoints to ensure functionality of the BCR while preventing autoreactivity [62]. B cells recognize large and soluble (not necessarily cell-bound) antigens including different macromolecules such as proteins, glycoproteins, and fatty acids [63]. The majority of mature B cells consist of FO B cells that recirculate between B-cell follicles in secondary lymphoid organs. FO B cells are predisposed to T-cell-dependent activation, a two-step process resulting in generation of plasma cells and memory B cells [64,65]. In a first so-called “extra-follicular” response, antigen activated B cells differentiate into short-lived plasmablasts. Plasmablasts are highly proliferative and provide early protection by production of antibodies (immunoglobulin M (IgM)) with relatively low affinity. With the help of T_FH_ cells, activated B cells form a germinal center. Here, B cells extensively proliferate and undergo affinity maturation, resulting in generation of plasma cells, which produce large quantities of high-affinity antibodies and memory B cells, which enable a specific immune response upon renewed antigen exposition after years. The maturation process involves isotype switching of immunoglobulins (from IgM to other isotypes such as IgG), modification of their antigen-binding sites by somatic hypermutation, and lastly, selection of B cells producing antibodies with the highest affinity [65]. MZ B cells mainly reside in the marginal zone of the spleen and provide a first, rapid defense against blood-borne microorganisms [66]. B1 cells, which reside in mucosal tissue and body cavities are long-lived, self-renewing B cells. They develop separately from FO and MZ B cells (also termed B2 cells) and are the main source of natural antibodies. These polyreactive antibodies are produced independent of previous antigen exposure and play an important role in protection from mucosal pathogens [67].

A few years after the first detection of antibodies in atherosclerotic lesions [26], immunoglobulins specifically recognizing epitopes of oxLDL were detected in plasma of diseased [68,69] and healthy [69] individuals, as well as in human plaques [70]. Subsequently, human oxLDL-reactive autoantibodies were shown to prevent oxLDL uptake by macrophages [71]. Dependent on the stage of atherosclerosis, macrophage uptake of oxLDL is proatherogenic [72,73]. Hence, immunoglobulins targeting oxLDL were suggested to be atheroprotective. Animal studies provided further evidence that B cell-related immunity can protect from atherosclerosis [74,75]. However, depletion of mature B cells using a monoclonal antibody against CD20 reduced atherosclerosis in mice [76]. The treatment decreased anti-oxLDL IgG titers while less affecting natural anti-oxLDL IgM antibodies. B cells are seen as a heterogenous population of distinct subsets with pro- or antiatherogenic properties. For a thorough review of the role of B cells in atherosclerosis, the reader is kindly referred to [77] and to the review from Binder et al. in this issue.

## 3. Principles of Vaccination

The term vaccination derives from *Variolae vaccinae* or “smallpox of the cow” and pays homage to the works of Edward Jenner, who in 1796 successfully inoculated a boy against smallpox [78]. Although the practice of immunization dates back even further, Jenner is considered the founder of immunization in the West [79]. In 1989, Charles Janeway Jr. proposed that a collection of receptors expressed by innate immune cells are responsible for detecting conserved products of microbial origin [80]. From this Nobel prize-winning theory came the concept of pattern recognition receptors (PRRs), which upon interaction with agonistic ligands activate innate immune cells [81]. This activation includes active microbicidal and proinflammatory mechanisms to eliminate infectious agents and the expression of surface co-stimulatory signals necessary for adaptive immune cells. There are four major subfamilies of PRRs: Toll-like receptors (TLRs), nucleotide-blinding oligomerization domain (NOD)-containing receptors (NLR), retinoic acid-inducible gene 1 (RIG-1)- like receptors, and C-type lectin receptors (CLRs). PRRs can recognize pathogen-associated molecular patterns (PAMPs) and damage-associated molecular patterns (DAMPs) [81]. While PAMPs drive inflammation in response to infections, DAMPs can be released by tissue stress and damage to induce a sterile inflammatory response, a concept brought forward by Polly Matzinger [82].

Whether by environmental exposure to a pathogenic agent or via immunization, mounting of a fulminant immune response begins through the activation of PRRs [83]. Opsonization of the pathogen’s surface by antibodies and other proteins aids in phagocytosis by APCs. Subsequently, pathogen-derived antigens are processed and presented in MHC class I or class II proteins on the APC surface. Under ideal conditions, vaccines should aim to trigger the innate immune system and a CD4, CD8, and B cell response. Antigen-activated CD4^+^ T cells are essential modulators of immune responses by the secretion of cytokines upon recognizing antigens, whereas the induction of cytotoxic CD8^+^ T cells is a critical mechanism of antiviral immunity. B cells also are important for antiviral immunity via secretion of antibodies that cover the surface of viruses to target them for degradation or prevent their intrusion into a host cell [83].

Several vaccination strategies are employed in the prevention of infection. Live attenuated vaccines, as exemplified by the milestone vaccination strategies against measles, mumps, and chickenpox, contain a laboratory-weakened version of the original pathogenic agent. Typically, live attenuated vaccines for viruses are far less tedious in manufacturing than for bacteria due to the smaller genome [78]. Inactivated vaccines, e.g. against influenza, are based on formulation of the destructed pathogenic agent with chemicals, heat, or radiation. These vaccines are more stable than live attenuated vaccines, do not need to be refrigerated, and are easier transported into rural areas. However, these vaccines produce a weaker immune response and require additional booster shots to maintain immunity. Toxoid vaccines are based on the toxin produced by a certain bacterium such as tetanus. The toxin is rendered harmless or toxoid and used together with proper adjuvants such as aluminum or calcium salts to elicit immunity [83]. Subunit vaccines, such as the recombinant hepatitis B vaccines, include only few antigenic parts of the pathogen and have a reduced likelihood of adverse reactions. Subunit vaccines can be further subdivided according to the antigenic species used: protein, polysaccharide, or conjugate. The utilization of a carrier protein or conjugate greatly induces a long-term protective response even in infants as opposed to plain polysaccharide vaccines [83].

Another strategy in conferring immunoprotection is via passive immunization. Passive immunization refers to the transfer of antibodies to an unprotected individual for prevention or treatment. This strategy was used in the mid-1890s with the diphtheria antitoxin, which drastically reduced mortality during diphtheria outbreaks. Mainly via providing immunoglobulins, passive immunization gives immediate, but short-lived protection, lasting several weeks to 4 months at most [83].

## 4. Vaccination as a Novel Strategy for the Treatment of Atherosclerosis

Today, vaccination research is expanding toward new targets in prevention and mitigation of autoimmune disorders and diseases, encompassing metabolic disorders, neurodegeneration, allergic disorders, and transplantation rejection. Recently, several experimental immunization strategies emerged as potential treatment options for atherosclerosis. 

### 4.1. Cholesterol-Lowering Vaccination Strategies

Proprotein convertase subtilisin/kexin 9 (PCSK9) is a liver-synthesized protein that plays a major regulatory role in cholesterol homeostasis. The clearance of LDL cholesterol is facilitated by binding to the LDL receptor (LDLR) expressed on hepatocytes [84]. Binding of LDL to LDLR causes internalization of the receptor, the release of LDL, and the recycling of the receptor to the surface. The binding of PCSK9 to LDLR blocks the conformational change of the hairpin necessary for surface reshuffling and leads to receptor degradation, thereby limiting the capacity to remove LDL particles from circulation [85]. PCSK9 inhibitors allow for more active receptors on the surface of hepatocytes and, therefore, increase clearance of LDL cholesterol.

PCSK9 inhibition has been investigated through various pharmacological mechanisms including monoclonal antibodies or formation prevention by antisense oligonucleotides and small interfering RNA (siRNA) [86]. Efficacy of three monoclonal PCSK-9 antibodies—evolocumab, alirocumab, and bococizumab—was tested in large clinical trials [87,88,89] (Table 1). Due to the occurrence of antidrug antibodies, bococizumab was discarded as clinical treatment strategy [89]. Evolocumab and alirocumab are used in clinical practice since 2015 and recently received class I recommendations as a second-line cholesterol-lowering treatment for high-risk patients in Europe [90]. Despite their high efficacy, frequent injections (at least monthly) and high costs maintain a current roadblock [91]. More recently, inclisiran, an siRNA-based drug preventing PCSK9 expression, proved beneficial in three phase III trials, enrolling more than 3000 patients with atherosclerotic cardiovascular disease (ORION-10 and ORION-11 trials) [92] and 482 patients with familial hypercholesterolemia (ORION-9) [93]. Inclisiran, currently awaiting approval in the USA and Europe, has a favorable drug regimen and needs to be administered only twice yearly. Several preclinical studies have substantiated the capability of active vaccines targeting PCSK9 to confer sustained cholesterol lowering [94,95,96] and consequently atheroprotection [96], mainly via induction of neutralizing antibodies. A phase I clinical trial was completed in 2017, but the results are yet to be published [97].

Another promising strategy is targeting apolipoprotein CIII (APOC3), a key regulator of plasma triglycerides and independent risk factor for cardiovascular disease. APOC3 can contribute to hypertriglyceridemia by inhibiting hydrolysis of triglyceride-rich lipoproteins [98] and reducing hepatic uptake of triglyceride-rich lipoproteins [99]. Individuals with reduced circulating APOC3 levels have a reduced risk of developing coronary heart disease [100,101] and administration of antisense DNA targeting APOC3 successfully reduced triglyceride levels [102]. Although a permanent reduction in APOC3 levels by a vaccination with virus-like particles successfully reduced plasma triglyceride levels in mice [103], the effects of an APOC3-targeting vaccine on atherosclerosis and cardiovascular disease remain unknown.

Other vaccination strategies aim to alter cholesterol esterification. In this process, cholesterol is converted to hydrophobic cholesteryl esters, which are linked to atherosclerosis. Cholesteryl ester transfer protein (CETP) is a plasma protein that facilitates transport of these cholesteryl esters and triglycerides between lipoproteins. Intramuscular immunization of New Zealand White rabbits, a species with naturally high CETP expression, with CETP in Freund’s complete adjuvant followed by immunizations of CETP in Freund’s incomplete adjuvant induced CETP-specific antibodies and hampered atherogenesis [104]. Furthermore, CETP inhibition was associated with high plasma HDL levels, substantiating that CETP transfers cholesteryl esters from antiatherogenic HDL to proatherogenic LDL [104]. The CETP inhibitor anacetrapib was previously tested in the phase III clinical trial REVEAL. The combinatory therapy of statins and the CETP inhibitor presented lower incidence of coronary heart disease and reduced non-high-density lipoprotein (non-HDL) cholesterol [105].

### 4.2. Active Immunization against Plaque-Associated Antigens

A great body of evidence suggests that an adaptive immune response against plaque-associated autoantigens is critically involved in atherogenesis [48,77] (Table 2). Preclinical studies have investigated the potential of immunization against such autoantigens to protect from atherosclerosis.

#### 4.2.1. Heat-Shock Proteins and Pathogens

HSPs are a family of highly conserved proteins abundantly produced by eukaryotic and prokaryotic cells under stressful conditions such as increased temperature [106]. They act as molecular chaperones, which aid in protein (re-)folding, thereby protecting proteins from denaturation or loss of function. Humans develop protective immunity against microbial HSPs after infection but can also mount immune responses to modified autologous HSPs. Traditional atherosclerotic risk factors, such as hypercholesterolemia, drive HSP60 expression in endothelial cells, which are subsequently targeted by pre-existing cellular and humoral immunity [106]. Detection of HSP60-specific IFN-γ-producing T_eff_ cells in early human atherosclerotic lesions identified HSP-related autoimmunity as a potential driver of atherogenesis [58]. Consequently, vaccination strategies to induce tolerogenic immunity against HSP have been investigated. Jing et al. reported an atheroprotective effect by oral immunization against mycobacterial HSP65 [107]. Concomitantly, HSP65-specific proliferation of splenocytes and IFN-γ production were suppressed, whereas IL-10 production was increased [107]. Additionally, sublingual immunization with recombinant HSP60 from *Porphyromonas gingivalis* (rGroEL) was found to provide significant atheroprotection in spontaneously hyperlipidemic (*Apoe*^shl^) mice, which was accompanied by a humoral response against rGroEL and an increase of FoxP3^+^ T_regs_ in submandibular glands [108].

#### 4.2.2. LDL, oxLDL, and ApoB

The most used antigens in preclinical active immunization strategies against atherosclerosis are LDL, oxLDL, and ApoB (see Table 2). Palinski and colleagues were the first to test the effect of immunization in atherogenesis by administering oxLDL to hypercholesterolemic rabbits [109]. Against former hypothesis, immunized rabbits exhibited reduced atherosclerotic lesions and elevated levels of antibodies against oxLDL [109]. Subsequent studies investigating LDL-based vaccination strategies confirmed atheroprotection in rabbits and mice [110,111,112,113]. Ameli et al. also demonstrated that immunization with native LDL reduced atherosclerosis in hypercholesterolemic rabbits [110]. As the two immunization regimens were associated with a similar increase of anti-oxLDL antibody levels, the authors hypothesized that the native LDL may have become oxidized during the immunization procedure. The lack of correlation between oxLDL antibody levels and lesion area suggested that immunization-mediated atheroprotection may have been predominantly driven by a cellular immune response [110]. This hypothesis was supported by immunization of LDL-receptor deficient (*Ldlr*^−/−^) mice with either oxLDL or native LDL. Both antigens reduced atherosclerosis to the same extent, but only the oxLDL-treated group had increased levels of anti-oxLDL IgG antibodies. These results implicate a cellular immune response in mediating antiatherogenic effects by immunization [113]. Indeed, T cells were critical for generation of oxLDL-specific antibodies and the reduction of atherosclerosis in oxLDL-immunized apolipoprotein E-deficient (*Apoe*^−/−^) mice [114]. These findings implicate that a successful vaccine targeting atherosclerosis needs to induce appropriate cellular and humoral immunity.

**Table 2 cells-09-02560-t002:** Experimental studies investigating immunization against LDL-related antigens in mice.

Year of Publication	Target Antigen(s)	Adjuvant/Carrier and Scheme	Administration Route	Mouse Model	Immune Response Experimentally Verified		Effect on Atherosclerosis	Reference
					Humoral	Cellular ^†^		
1998	MDA-LDL	1× CFA/ 5× IFA	s.c.	*Apoe* ^−/−^	+	−	53% ↓	George et al. [112]
1998	1. MDA-LDL 2. Native LDL	1× CFA/ 7× IFA	1× s.c. + 7× i.p.	*Ldlr* ^−/−^	1. IgG ^a^	−	1. 46% ↓ 2. 37% ↓	Freigang et al. [113]
2001	1. MDA-LDL 2. PH	1× CFA/ 4× IFA	food pad injection	*Apoe* ^−/−^	IgG	+	1. 39% ↓ 2. 46% ↓	Zhou et al. [114]
2003	p143 + p210	2× Alum + cationized BSA	injection	*Apoe* ^−/−^	IgG	−	60% ↓	Fredrikson et al. [115]
2004	Native LDL	4× IL-12	1× s.c., 3× i.p.	*Apoe* ^−/−^	IgG ^a^	+	↓ ^b^	Chyu et al. [116]
2005	1. MDA-p45 2. MDA-p74	3× Alum + cBSA	injection	*Apoe* ^−/−^	1. IgG1 2. IgG1 + IgM	+	1. 48% ↓ 2. 31% ↓	Fredrikson et al. [117]
2005	MDA-LDL	1× CFA/5× IFA	s.c.	*1. Apoe*^−/−^ *2. CD4*^+^/ *Apoe* DKO ^c^	1.IgG1/2a/2b, IgM 2. IgG1/2a/2b	−	1. 39% ↓ 2. 41% ↓	Zhou et al. [118]
2005	1. p1 2. p2 ^d^	2× Alum	1× s.c., 1× i.p.	*Apoe^−^* ^/−^	IgM + IgG	+	2. 40% ↓ ^e^	Chyu et al. [18]
2006	1. MDA-LDL 2. oxLDL ^d^	4×, none	p.o.	*Ldlr* ^−/−^	−	+++	2. 71% ↓	van Puijvelde et al. [119]
2007	PC	9× 1826 CpG oligonucleotide	i.p.	*Apoe* ^−/−^	IgM + IgG	+	40% ↓	Caligiuri et al. [120]
2008	1. p45 2. p210	3× Alum + cBSA	injection	*Ldlr* ^−/−^ *huB100^tg^*	IgM	−	1. 66% ↓ 2. 59% ↓	Fredrikson et al. [19]
2010	oxLDL	3× dendritic cells ^f^	i.v.	*Ldlr* ^−/−^	IgG1/2c	+	87% ↓	Habets et al. [121]
2010	p210	24× cholera toxin B	i.n.	*Apoe* ^−/−^	IgG	+++	35% ↓	Klingenberg et al. [122]
2010	oxLDL	1× CFA/1× IFA	s.c.	*huB100^tg^*	−	+	NA	Hermansson et al. [53]
2011	1. ApoB +HSP 60 2. ApoB 3. HSP60	5× CFA+RIBI adjuvant	s.c.	*Ldlr* ^−/−^	IgG + IgM	++	1. 41% ↓ 2. 15% ↓ 3. 21% ↓	Lu et al. [123]
2011	ApoB100	Tolerogenic dendritic cells ^g^	i.v.	*Ldlr* ^−/−^ *huB100^tg^*	IgG	++	70% ↓	Hermansson et al. [124]
2011	p210	4× Alum + cBSA	s.c.	*Apoe* ^−/−^	−	++	37% ↓ ^d, h^	Wigren et al. [125]
2012	p210	3× Alum + cBSA	s.c.	*Apoe* ^−/−^	−	+	57% ↓	Chyu et al. [126]
2012	1. p210+p240+MDA-p210 2. p210	14 days, none	s.c. (mini-pumps)	*Apoe* ^−/−^	−	+++	1. 40% ↓ ^h^ 2. 30% ↓ ^h^	Herbin et al. [127]
2012	oxLDL ^d^	5×, none	i.n.	*Apoe* ^−/−^	−	++	48% ↓ ^i^	Zhong et al. [128]
2013	1. HSP60 ^j^ + p45 2. HSP60 3. p45	5× KLH	p.o.	*Ldlr* ^−/−^	−	++	1. 40% ↓ ^d^ 2. 27% ↓ 3. 29% ↓	Mundkur et al. [129]
2013	1. p3 ^k^ 2. p6 ^k^	1× CFA/4× IFA	1x s.c., 4x i.p.	*Apoe* ^−/−^	IgG1/2c	+	40% ↓	Tse et al. [54]
2017	1. p1 2. p2 3. p ^k^ 4. p4 5. p5 6. p6 ^k, l^	2× CFA supplemented with Mycobacterium tuberculosis H37Ra	s.c.	*Apoe* ^−/−^	−	3/6 +	6 ↑	Shaw et al. [130]
2017	1. p101 ^k^ 2. p102 ^k^ 3. p103 ^k^	1× CFA/4× IFA	1× s.c., 4× i.p.	*Apoe* ^−/−^	IgG1/2c	++	1. 39% ↓ 2. 37% ↓ 3. 40% ↓	Kimura et al. [17]
2018	MDA-p210	1× CFA/4× IFA	s.c.	*Apoe* ^−/−^	+	−	46% ↓	Zeng et al. [131]
2018	p6 ^k^	1. 6× Addava× 2. 1× CFA/5× IFA	1. s.c. 2. 1× s.c., 5× i.p.	*Apoe* ^−/−^	2. IgG1/2c	1. +	50% ↓	Kobiyama et al. [132]
2018	p18 ^k, m^	1× CFA/4× IFA	1× s.c., 4× i.p.	*Apoe* ^−/−^	IgG1/2c	+++	35% ↓	Kimura et al. [16]
2018	p3500 ^n^	4× DSPG-liposomes	i.p.	*Ldlr* ^−/−^	−	+	50% ↓	Benne et al. [133]
2018	ApoB100	1× CFA/ 1× IFA	s.c.	*Ldlr* ^−/−^ *huB100^tg^*	IgG1/2c	+	↓	Gisterå et al. [134]

**^†^** +++, evidence of an ApoB-specific T_reg_ response; ++, evidence of a T_reg_ response; +, indirect evidence of tolerogenic cellular immune response (e.g., increased levels of anti-inflammatory cytokines) or cellular immune response not related to T_regs_; −, no cellular immune response ascertained; ^a^ correlation with atherosclerosis; ^b^ atheroprotective when 6–7 week old mice were immunized, but not when immunized at 20 weeks of age; ^c^ 70% atherosclerosis reduction + reduced anti-MDA-LDL IgG1/2a/2b and IgM in DKO vs. ApoE^−/−^ mice; ^d^ delayed immunization, administered to older mice with already established atherosclerosis, also conferred atheroprotection; ^e^ effect absent in splenectomized mice; ^f^ bone-marrow-derived dendritic cells, matured in vitro and pulsed with oxLDL; ^g^ bone-marrow-derived dendritic cells, matured in vitro and pulsed with ApoB + IL-10; ^h^ atheroprotective effect inhibited by administration of T_reg_ depleting (anti-CD25) antibodies; ^i^ atheroprotection partly inhibited by administration of anti-TGF-β antibody; ^j^ peptide 153–163; ^k^ proven to bind murine MHC-II; ^l^ immunization after pre-feeding a Western-type diet for 5 weeks; ^m^ proven to bind human MHC-II; ^n^ predicted to bind murine MHC-II. Abbreviations: ApoB = apolipoprotein B, *Apoe*^−/−^ = apolipoprotein E-deficient, cBSA = cationized bovine serum albumin, CFA = Freund’s complete adjuvant, DC = dendritic cell, DKO = double knockout, DSPG = 1,2-distearoyl-*sn*-glycero-3-phosphoglycerol, HSP = heat-shock protein, *huB100^tg^* = human ApoB transgenic, IFA = Freund’s incomplete adjuvant, IgG = immunoglobulin G, IgM = immunoglobulin M, i.n. = intranasal, i.p. = intraperitoneal, i.v. = intravenous, KLH = keyhole limpet hemocyanin, LDL = low-density lipoprotein, *Ldlr*^−/−^ = LDL receptor-deficient, MDA = malondialdehyde, oxLDL = oxidized LDL, PC = phosphocholin, PH = plaque homogenate, p.o. = per os, s.c. = subcutaneous, T_regs_ = regulatory T cells.

A prerequisite for the development of an effective and clinically feasible vaccination is the identification of immunogenic epitopes. For this purpose, Fredrikson et al. generated a library of overlapping ApoB peptides covering the whole sequence of human ApoB-100 and identified 102 epitopes that were recognized by human antibodies [135]. Many of these peptides were recognized in their native state, as well as after modification with malondialdehyde (MDA), a process that occurs during oxidation of LDL [135]. Immunization with some of these peptides, such as native p143, p210 [115], and p2 [18] or MDA-modified p45 and p74 [117] reduced atherosclerosis in *Apoe*^−/−^ mice, whereas vaccination with p1 did not yield any beneficial effect [18]. Subsequent work substantiated the critical role of cellular immunity in vaccine-induced atheroprotection [16,17,19,119,122,125,127]. Firstly, immunization with ApoB peptides (p45 and p210) reduced atherosclerosis in *Ldlr*^−/−^ mice expressing human ApoB-100 without inducing peptide-specific IgG antibodies [19]. Secondly, the atheroprotective effect observed in *Apoe*^−/−^ mice immunized intranasally with p210 was associated with an expansion of antigen-specific T_regs_ in the spleen, which effectively suppressed an effector T-cell response to the respective ApoB peptide in vitro [122]. Oral administration of oxLDL to *Ldlr*^−/−^ mice induced expansion of T_regs_ in secondary lymphoid organs and protected from atherosclerosis [119]. In another study, adoptive transfer of tolerogenic DCs pulsed with human ApoB in presence of IL-10 into *Ldlr*^−/−^ mice expressing human ApoB-100 significantly reduced atherosclerosis [124]. Given that these DCs induced T_regs_ in vitro, the observed atheroprotection was attributed to the induction of a T_reg_ response, although in vivo proof was missing [124]. Lastly, depletion of T_regs_ prevented atheroprotection in *Apoe*^−/−^ mice immunized with ApoB peptides, substantiating that a T_reg_ response is critically involved in mediating the antiatherogenic effect of a vaccine [125,127].

Of note, immunization with p210 elicited an antigen-specific antibody response and a T_reg_ response [115], which was critically involved in mediating the atheroprotective effect of the vaccine [125,127]. Surprisingly, p210 has a biologically nonrelevant binding affinity to the MHC-II allele (I-A^b^) expressed by the mouse models used, suggesting that p210 does not confer atheroprotection by presentation in MHC-II complexes to T_regs_ [136]. To account for MHC-II restriction of CD4^+^ T-cell epitopes, the binding affinity of murine ApoB peptides to I-A^b^ was modeled and subsequently confirmed by a competitive binding assay [17,54]. *Apoe*^−/−^ mice immunized with peptides identified using this approach (15-mers p3, p6 [54], and p18 [16]; 16-mers p101, p102 and p103 [17]) exhibited a significant reduction in lesion size concomitant with an expansion of IL-10-producing peritoneal T_regs_ and relevant antibody titers to the peptides [16,17,54]. The peptide p18 sequence is identical in mice and humans and binds to murine I-A^b^ and several human MHC-II alleles [16]. Using an innovative MHC-II-p18 tetramer (a fluorochrome-labeled complex consisting of four recombinant MHC-II molecules loaded with p18), p18-specific CD4^+^ T cells were for the first time directly detected by flow cytometry in mouse and human samples. Utilization of these tetramers demonstrated significant expansion of p18-specific T_regs_ upon vaccination [16].

Recently, Gisterå et al. substantiated the critical role of a humoral response to LDL in *Ldlr*^−/−^ mice which expressed human ApoB and had transgenic, human ApoB-reactive T cells [134]. Some of the transgenic T cells survived thymic selection, mainly differentiated into T_FH_ cells, and enabled production of anti-LDL IgG antibodies, which promoted LDL clearance from the circulation and led to a significant reduction of atherosclerosis [134]. Taken together, there is now profound evidence that antigen-specific regulatory CD4^+^ T cells and antibodies are critically involved in mediating the atheroprotective effect of ApoB-based vaccination strategies (Figure 1). Outside of the CD4^+^ T cell lineage, a specific CD8^+^ T cell subset with suppressive capacities was identified to mediate atheroprotection upon immunization with ApoB peptide p210 [126,137].

#### 4.2.3. Multitarget Vaccines

Multitarget vaccines are mixtures of individual vaccines. These can target one particular disease-inducing pathogen, which has multiple serotypes, or a combination of multiple pathogens, such as the diphtheria, tetanus, and pertussis (DTP) vaccination [138]. Lu et al. administered multi-antigenic epitopes from ApoB (p45), HSP60 (peptides 153–163/303–312), and *Chlamydophila pneumoniae* complexed in a dendroaspin scaffold with alum as adjuvant to *Ldlr*^−/−^ mice [139]. This multitarget vaccine induced a stronger T_reg_ response and conferred significantly enhanced atheroprotection compared to single or bi-epitope controls [139]. Combining antigenic epitopes in a vaccine could offer an opportunity to increase efficacy of antiatherosclerotic immunization strategies.

In summary, several approaches of immunizations with LDL, oxLDL, or ApoB-related peptides yielded substantial atheroprotection. Depending on the specific vaccination protocol and the antigens used, the adaptive immune response conferring the observed atheroprotective effects may induce regulatory CD4^+^ T cells, suppressive CD8^+^ T cells, T_FH_, and anti-LDL antibodies. Together, these promising findings encourage future studies to more precisely dissect the adaptive immune mechanisms involved in atherosclerosis to design a precise therapy combating this deadly disease.

## 5. Challenges in Translating an Antiatherosclerotic Vaccine into Clinical Practice

Preclinical studies have underlined the therapeutic potential of an antiatherosclerotic vaccination strategy. However, before such implementation into clinical practice, several obstacles will have to be overcome.

### 5.1. Identification of Optimal Epitopes

A vaccine aiming to induce a CD4^+^ T-cell response requires presentation of the respective peptide epitopes by MHC-II, which is called the human leukocyte antigen (HLA) complex in humans. By combining a prediction algorithm and competitive binding assays, the ApoB peptide p18 was identified to bind the HLA-DR allele DRB1*07:01 [16], which is relatively common variant with a frequency of 8–14% dependent on race [140]. Yet, the vast heterogeneity of HLA types and alleles [140] presents an obstacle in translating a vaccine to clinical practice.

The ideal scenario represents a single epitope capable of binding all HLA types. However, not all ApoB peptides bind all HLA types with similar affinity. A broadly applicable vaccine should, therefore, comprise multiple immunogenic ApoB peptides on the basis of HLA testing. Recently, Wolf et al. compiled a list of 30 ApoB peptides binding broadly different HLA types using a computer algorithm-based prediction and in vitro affinity measurement [55]. The peptide pool successfully induced in vitro responses of T cells isolated from healthy volunteers and patients with coronary artery disease [55]. The translation of this experiment into a mouse model is difficult and has not yet been performed. First, only the ApoB peptide p18 shares sequence homology between mice and humans [16], while the other peptides could induce a “foreign” adaptive immune response, which could interfere with the outcome and interpretation of the experiment. Second, although individual human HLA-transgenic mice have been described, it would require substantially more transgenic mouse strains backcrossed to an atherosclerosis-susceptible background.

An alternative promising approach already successfully tested in clinical trials of cancer vaccines is the Ii-Key hybrid technology [141]. Ii-Key is a fragment of the Ii protein (or MHC class II-associated invariant chain), which physiologically binds to MHC-II molecules after synthesis, preventing binding to endogenous peptides. It is removed before binding of exogenous peptides. Ii-key binds all HLA-types with high affinity. Linking MHC-II-restricted epitopes to Ii-key enhances their potency to active CD4^+^ T cells and, most notably, does not require HLA matching, thereby enabling the use of a unique epitope for all individuals. However, this technology has not been used in atherosclerosis.

### 5.2. Design and Selection of Effective and Clinically Applicable Adjuvants

Adjuvants are critical components of vaccines that augment the immune response to the target antigen by activating the innate immune system [142]. Adjuvants need to be carefully selected depending on several parameters, such as the antigen used, desired immune response, and administration route [142]. Adjuvants commonly used in preclinical studies investigating antiatherosclerotic vaccine strategies include complete or incomplete Freund’s adjuvant (CFA or IFA), aluminum, and cholera toxin B subunit (CTB; Table 2). Whereas aluminum has been used in clinical practice for almost a century and CTB has been successfully tested in clinical trials, CFA is not clinically applicable due to potential toxic side effects [143]. Although IFA is still being studied in clinical trials of human immunodeficiency virus (HIV) and cancer vaccines, its use was largely discontinued due to safety concerns [144].

Recently, Kobiyama et al. compared the potential of six clinically applicable adjuvants formulated individually with the ApoB peptide p6 to protect from atherosclerosis [132]. Of these, Addavax, a squalene-based adjuvant similar to MF59, which is approved for an influenza vaccine in several countries including the USA [145], yielded the best results. It conferred a similar atheroprotection observed in *Apoe*^−/−^ mice vaccinated with CFA/IFA (~50% lesion reduction) [132]. The vaccination induced IL-10 production in restimulated cells, but did not induce an antigen-specific antibody response. Although the exact mechanism of protection is not completely understood, Addavax emerges as a promising candidate for clinical translation of a vaccine against atherosclerosis.

Another study demonstrated the feasibility of an adjuvant-free immunization with ApoB peptides [127]. Continuous delivery of a low-dose ApoB–peptide mix (p210, p240, and MDA-p210) without adjuvant over 2 weeks via subcutaneously implanted minipumps significantly reduced atherosclerosis in *Apoe*^−/−^ mice through induction of a T_reg_ response [127].

Lastly, administration of ApoB peptides encapsulated in liposomes was revealed to confer atheroprotection and induced a T_reg_ response specific to the encapsulated antigen [133]. Given that liposomes have been extensively used as drug delivery systems in clinical practice since discovery in 1965 [146], this approach represents an additional encouraging option for realization of an antiatherosclerotic vaccine.

### 5.3. Ensuring Stability of the Atheroprotective Immune Response

Vaccination with MHC-II-restricted ApoB peptides has been shown to be mediated by an expansion of ApoB^+^ T_regs_ [16,17]. Yet, the development of atherosclerosis in *Apoe*^−/−^ mice is accompanied by a conversion of T_regs_ into proatherogenic T_FH_ cells [147] or dysfunctional IFN-γ-producing T_H_1/T_reg_ cells with reduced suppressive capacities [148,149]. A recent study demonstrated that, under hypercholesterolemic conditions, ApoB^+^ T cells are especially prone to switching their phenotypes [55]. Particularly, these cells were shown to lose expression of the T_reg_ defining TF FoxP3 and to acquire a T_H_1/T_H_17 proatherogenic phenotype. Adoptive transfer of ApoB^+^ T_regs_ did not protect *Apoe*^−/−^ mice from atherosclerosis progression, as the transferred cells likely lost their atheroprotective T_reg_ phenotype [55]. Similarly, ApoB^+^ T cells isolated from blood of atherosclerotic patients predominantly exhibited proatherogenic phenotypes [16,55]. We refer the interested reader to the review by Wolf et al. in this issue for deeper insight into ApoB^+^ T-cell responses in atherosclerosis.

The specific mechanisms underlying the deleterious T cell conversion in atherosclerosis are largely unknown. Clinical translation of a T_reg_-based vaccine strategy requires a fundamental understanding of the switch underlying mechanisms. Vaccination could cause serious harm when T_regs_ are induced, which would subsequently convert into disease-promoting effector T cells.

### 5.4. Determining an Optimal Vaccination Scheme and Administration Route

In most preclinical studies, vaccinations were administered to young mice before exposing them to a cholesterol-enriched, Western-type diet (WD) (Table 2). The vaccination protocols predominantly followed a preventive approach aiming to induce atheroprotective immunity before disease onset. A comparable approach in clinical practice would imply vaccinating children or teenagers that have not yet faced an unhealthy lifestyle with potentially harmful effects on their responsiveness to immunization. Although such a strategy is well established for infectious disease prevention, it would be of particular interest to develop a vaccination applicable for tertiary prevention, i.e., for the treatment of patients who have already developed atherosclerosis. Considering the T_reg_ switch in atherosclerosis, immunization of patients with established disease has to be carefully assessed.

Preclinical studies showed that the timing of vaccination is important. Immunization of 6–7-week old *Apoe*^−/−^ mice conferred atheroprotection, but yielded no significant benefit in 20 week old mice [116]. Moreover, immunization against the ApoB-peptide p6 was atheroprotective in 8-week old *Apoe*^−/−^ mice [54], but enhanced atherosclerosis in 13-week old *Apoe*^−/−^ mice pre-fed a WD for 5 weeks [130]. Of note, this proatherogenic effect was revealed to be epitope-specific; immunization against another ApoB peptide, p3, using the same vaccination protocol did not accelerate atherogenesis [130]. Yet, in some other studies, delayed immunization regimens [18,119,127,128] were shown to confer atheroprotection. Apart from these few examples, studies directly comparing the efficacy of different antiatherosclerotic vaccination schemes (e.g., different dosages or administration frequencies) are largely lacking. Many vaccines against infectious agents require periodical administration to maintain protective immunity. Whether repetitive applications of antiatherosclerotic vaccines over an extended timespan might be an option to improve stability of the atheroprotective immune response has to be determined.

Antiatherosclerotic vaccines have been successfully applied nasally, orally, subcutaneously, intraperitoneally, or intramuscularly (Table 2). Of these, subcutaneous and intraperitoneal injections were most commonly used, and they robustly induced atheroprotective humoral and cellular immune responses. Mucosal administration of low antigen doses is generally regarded as a very effective way of inducing tolerogenic T_regs_ [150]. Accordingly, oral [20,22,129,151] or nasal [122,128,152] administration of HSP- or LDL-related antigens confers atheroprotection. Studies directly comparing the efficacy of mucosal or parenteral application of a distinct antigen to prevent or treat atherosclerosis have not yet been performed. Thus, the optimal administration route for an antiatherosclerotic vaccination strategy remains to be determined. Subcutaneous, intramuscular, and mucosal administration routes would principally be feasible for humans.

### 5.5. Development of New Markers for Patient Selection and Monitoring Treatment Response

After identification of a safe delivery method for an antigen and adjuvant, which confers long-term atheroprotection in mice, the next step includes treatment of large animal models, which has yet to be conducted. A subsequent clinical trial would consist of treating a selected high-risk group. Nevertheless, open questions involve how to define and assess risk. Given that immunization modulates the adaptive immune response, it would be particularly useful to identify new risk factors that promote or specifically accompany the development and progression of atherosclerosis and are not effectively targeted by existing treatments.

Tools to closely monitor treatment responses are a prerequisite for clinical testing. CANTOS and other recent trials used the acute-phase protein hsCRP to measure the inflammatory risk [10], which reliably predicted cardiovascular events in healthy individuals [153]. Although hsCRP might be a helpful parameter in risk stratification, its levels increase due to inflammatory responses and infections. Identification of more specific markers reflective of adaptive immune responses are highly desirable. In clinical practice, specific autoantibodies are used to diagnose and monitor several autoimmune diseases achieving a specificity of up to 100% [154]. Some studies have reported associations between presence of antibodies against several atherosclerosis-related autoantigens and overall disease burden or clinical outcome. However, the findings were inconsistent [77]. Recent evidence suggests that autoantibodies against LDL-related antigens (such as ApoB) are not harmful but rather atheroprotective, which might explain the inconsistent findings of clinical studies [77]. Given their established role as critical modulators of atherogenesis, ApoB^+^ T cells might be utilized as valuable risk markers. Low numbers of circulating T_regs_ have been shown to predict cardiovascular events in healthy individuals [155] as well as in patients with pre-existing CVDs [156]. However, other studies have not found an association between T_reg_ numbers and cardiovascular risk [157,158]. None of these studies tested the antigen specificity of the assessed T cells [48]. MHC-II–ApoB–peptide tetramers and restimulation assays now enable analysis of ApoB^+^ T cells [16,55]. Whereas the clinical feasibility of such approaches is yet limited by high costs, technological expertise, the requirement of HLA-matching for tetramers, and time intensiveness, they form the basis for development of novel risk prediction tools.

## 6. Clinical Trials Using Immunomodulatory Strategies to Treat Atherosclerosis

Active immunization strategies for the treatment of atherosclerosis have yet to be tested in clinical studies. However, immune modulatory therapeutic trials are currently underway. Here, we elaborate further on notable clinical trials that aim to unveil atherogenic pathology.

### 6.1. The LILACS Trial

A current on-going investigation is the LILACS (low-dose interleukin-2 in patients with stable ischemic heart disease and acute coronary syndromes) trial. Here, a human recombinant form of IL-2, aldesleukin (Proleukin^®^, Novartis), is administered in varying low-dose strategies to patients with stable ischemic heart disease and acute coronary syndromes [159]. IL-2 was reported to share a critical role in T_reg_ development, expansion, survival, and improvement of suppressive functionality [160,161]. Clinical studies in graft-versus-host disease [162], type 1 diabetes [163], and hepatitis C virus-induced vasculitis [164] reported that low-dose IL-2 supplementation could selectively promote T_reg_ expansion and suppress proinflammatory immune responses.

### 6.2. The GLACIER Trial

The GLACIER (Goal of Oxidized LDL and Activated Macrophage Inhibition by Exposure to Recombinant Antibody) trial, a phase II clinical study, assessed the effect of an anti-oxLDL targeted monoclonal antibody—MLDL1278A—on hindering atherosclerotic inflammation [165]. No significant effect on plaque inflammation assayed by positron emission tomography (PET) imaging was observed 12 weeks post antibody therapy. One discussed speculation as to why the trial did not meet its primary goal is that in contrast to the CANTOS trial inclusion criteria for participants did not have a cutoff for hsCRP levels. Additional factors that contributed to a neutral outcome could have included incorrect selection of biomarkers, a too short time window of a follow-up PET scan, a lack of coronary artery assessment, and the antibody not making it inside of the plaque.

### 6.3. Clinical Trials on Infectious Agents

To date, many studies have found implications that viral and bacterial pathogens contribute to atherosclerotic disease progression [166]. *Chlamydia pneumonia* has been reported to accelerate lesion progression in mouse [167] and rabbit [168,169] models of atherosclerosis. This led to testing antibiotic therapy in the prevention of coronary heart disease progression [170]. Four large-scale clinical trials, WIZARD, ACES, CLARICOR, and PROVE-IT-TIMI, collectively enrolled 20,000 patients. The trials tested either azithromycin (a macrolide antibiotic) or gatifloxacin (a fourth-generation fluoroquinolone antibiotic) with follow-up between 3 months and 2 years [171,172,173,174]. However, no long-term benefits arose from such treatments for patients with coronary heart disease or acute coronary syndromes excluding antibiotics as potential treatment against CVD.

## 7. Conclusions

Since immunoglobulins were first detected in atherosclerotic plaques four decades ago, a substantial body of evidence has further underlined the implications of adaptive immunity in atherogenesis. Given its high global prevalence and mortality, implementation of new immunomodulatory therapeutic strategies beyond lipid control is critical. Immunization regimens against plaque-related autoantigens display a promising approach. Such a vaccination strategy with LDL-derived peptides ideally induces antibodies that promote clearance of LDL cholesterol, as well as other atherogenic particles, and generates a T_reg_ response, which specifically suppresses atherosclerosis-relevant proinflammatory T-cell responses.

In particular, ApoB-related peptides have emerged as promising candidate antigens. ApoB-related epitopes binding to human MHC-II complexes and circulating T cells specifically reacting to these have been detected. Multiple preclinical studies demonstrated successful atheroprotection upon immunization with ApoB peptides, which could be related to expansion of antigen-specific T_regs_ and protective antibody responses. Despite these promising findings, several challenges will have to be taken before a vaccine against atherosclerosis is translated into clinical practice. These include gaining better insight into the underlying immune response, designing an optimal vaccine formulation comprising broadly effective but tolerable adjuvants and epitopes, epitope mixes, or epitope hybrids, and identifying the most successful application route. Furthermore, mechanisms underlying the phenotype conversion of T cells must be clarified to develop strategies preventing this switch. Lastly, different vaccine schemes must be evaluated for their efficacy, and new biomarkers for patient selection, as well as treatment monitoring, will have to be discovered. Overcoming these issues will require substantial efforts. However, the idea of an antiatherosclerotic vaccine seems to be in reach and could dramatically improve the treatment of the deadliest disease in the world.

## Figures and Tables

**Figure 1 cells-09-02560-f001:**
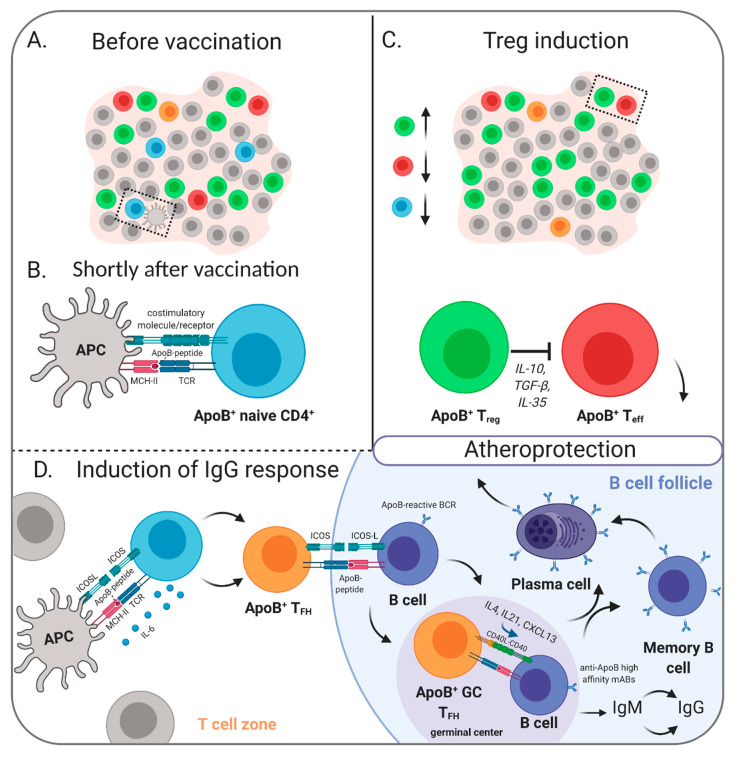
Proposed mechanisms of ApoB-related vaccines. (**A)** Before vaccination, naïve and differentiated ApoB-reactive (ApoB^+^) CD4^+^ T cells exist. In healthy individuals, these mainly comprise T_regs_. (**B**) Shortly after vaccination, antigen-presenting cells (APCs) present ApoB-peptides in MHC-II molecules to cognate naïve CD4^+^ T cells, which become activated. Activated CD4^+^ T cells proliferate and differentiate into effector subsets (T_eff_) including regulatory (T_regs_) and follicular helper (T_FH_) cells. The differentiation is dependent on the cytokine milieu and costimulatory signals. (**C**) T_regs_ confer atheroprotection through various mechanisms including secretion of anti-inflammatory cytokines and inhibition of ApoB^+^ T_eff_ cells. (**D**) T_FH_ cells aid in differentiation of B cells into plasma cells and memory B cells. Plasma cells produce high-affinity anti-ApoB IgG antibodies that confer atheroprotection via promotion of LDL clearance. Created with BioRender.com.

**Table 1 cells-09-02560-t001:** Clinical trials testing proprotein convertase subtilisin/kexin 9 (PCSK9) antibodies.

Trial	Antibody	*N*	Median Follow-Up	Main Inclusion Criteria	Dosing Scheme	Main Finding
FOURIER [87]	Evolocumab	27,564	2.2 years	ACSVD and high CVR, LDL-C ≥70 mg/dL or non-HDL-C ≥100 mg/dL despite high-intensity statin treatment ^1^ with/without ezetimibe	140 mg every 2 weeks or 420 mg monthly	Significant reduction of the primary endpoint ^2^ (RRR 15%) and key secondary endpoint ^3^ (RRR 20%)
ODYSSEY [88]	Alirocumab	18,924	2.8 years	ACSVD with ACS 1–12 months earlier, LDL-C ≥70 mg/dL or non-HDL-C ≥100 mg/dL or ApoB ≥80 mg/dL despite high-intensity statin treatment ^4^	75 mg every 2 weeks ^5^	Significant reduction of the primary endpoint ^6^ (RRR 15%) and all-cause mortality (RRR 15%)
SPIRE 1/2 [89]	Bococizumab	27,438	10 months ^7^	Previous CVE or high CVR ^8^, LDL-C ≥70/100 mg/dL or non-HDL-C ≥100/130 mg/dL (SPIRE 1/2) despite high-intensity statin treatment ^9^	150 mg every 2 weeks ^10^	No significant reduction of the primary endpoint ^11^

^1^ At least atorvastatin 20 mg daily; ^2^ composite of cardiovascular death, myocardial infarction, stroke, hospitalization for unstable angina, or coronary revascularization; ^3^ composite of cardiovascular death, myocardial infarction, or stroke; ^4^ atorvastatin 40–80 mg once daily, rosuvastatin 20–40 mg once daily, or the maximum tolerated dose of one of these statins (including no statin in the case of documented unacceptable side effects); ^5^ dose was adjusted under blinded conditions to target a LDL cholesterol level of 25–50 mg/dL and to avoid sustained levels below 15 mg/dL; ^6^ death from coronary heart disease, nonfatal myocardial infarction, fatal or nonfatal ischemic stroke, or unstable angina requiring hospitalization; ^7^ the trial was stopped prematurely, since high rates of antidrug antibodies occurred and the sponsor discontinued development of bococizumab; ^8^ history of diabetes, chronic kidney disease, peripheral vascular disease with additional cardiovascular risk conditions, familial hypercholesterolemia, and at least one additional cardiovascular risk factor; ^9^ atorvastatin ≥40 mg daily, rosuvastatin ≥20 mg daily, or simvastatin ≥40 mg daily, unless patients could not take those doses without side effects; ^10^ dose was reduced to target an LDL cholesterol level of ≥10 mg/dL; ^11^ nonfatal myocardial infarction, nonfatal stroke, hospitalization for unstable angina requiring urgent revascularization, or cardiovascular death. Abbreviations: ACS = acute coronary syndrome (myocardial infarction or unstable angina), ACSVD = atherosclerotic cardiovascular disease, ApoB = apolipoprotein B level, CVE = cardiovascular event, CVR = cardiovascular risk, LDL-C = low-density lipoprotein cholesterol level, non-HDL-C = non-high-density lipoprotein cholesterol level, RRR = relative risk reduction.

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
