# Peer review of "Vaccination in Atherosclerosis"

_cells, 2020, doi:10.3390/cells9122560_

Round 1
Reviewer 1 Report
The article, denominated Vaccination in Atherosclerosis, is a review that describes vaccine strategies to treated atherosclerosis.
The article describes the main strategies used to produce vaccines and the ones that are in course to treated CVD. The authors made a good bibliographic review of the subject and described the main point about the production of a vaccine.
specif comments:
line 239, aside from the article description, the author could add a table with the main tested antibodies and their results, resuming the text.
Line 269, the author could the explanation about CETP, indicating the model it was studied.
In line 271, a great body of evidence.... The author could mention some references to this evidence
The article could contain an illustrated conclusion, or a table that summarized the contents
Author Response
The article, denominated Vaccination in Atherosclerosis, is a review that describes vaccine strategies to treated atherosclerosis.
The article describes the main strategies used to produce vaccines and the ones that are in course to treated CVD. The authors made a good bibliographic review of the subject and described the main point about the production of a vaccine.
specif comments:
- line 239, aside from the article description, the author could add a table with the main tested antibodies and their results, resuming the text.
- Line 269, the author could the explanation about CETP, indicating the model it was studied.
- In line 271, a great body of evidence.... The author could mention some references to this evidence
- The article could contain an illustrated conclusion, or a table that summarized the contents
We thank the reviewer for the positive comments.
- We have now included a table highlighting the three major Phase III clinical trials evaluating efficacy of anti-PCSK9 antibodies (Page 6 line 280-300) and included short reference in the text (p.6 lines 255-257)
“Efficacy of three monoclonal PCSK-9 antibodies - Evolocumab, Alirocumab, and Bococizumab - has been tested in large clinical trials [87–89] (Table 1). Due to the occurrence of antidrug antibodies, Bococizumab was discarded as clinical treatment strategy [89].”
- We have now elaborated on the model that was utilized in the respective study:
Page 7 line 305-308:
“Intramuscular immunization of New Zealand White rabbits, a species with naturally high CETP expression, with CETP in Freund’s complete adjuvant followed by immunizations of CETP in Freund’s incomplete adjuvant induced CETP-specific antibodies and hampered atherogenesis”
- We have delineated the role of adaptive immunity in atherogenesis in detail in section 2, citing several references to support this statement and refer to table 2.
Page 6 lines 312-313:
“A great body of evidence suggests that an adaptive immune response against plaque-associated autoantigens is critically involved in atherogenesis [45, 72] (Table 2).”
- Thank you for this suggestion. We think that the figure and tables already included summarize the most critical content of this review and would like to leave it to the discretion of the editor to use the provided figure as a graphical abstract to summarize the manuscript.
Reviewer 2 Report
The review article "Vaccination in Atherosclerosis" not only describes possible approaches to solving the problem of atherosclerosis, but also reveals a number of unsolved problems of creation, specification, administration of vaccines, as well as a whole spectrum of still unidentified and ambiguous cellular interactions arising during the formation of an immune response. In my opinion, the important points of the article are that the authors emphasize the autoimmune component in the development of atherosclerosis.
The parts of the review article devoted to T- and B-cells look protracted, perhaps it is worth removing the well-known facts about cell differentiation and homeostasis, since this is not directly related to the topic of the review article.
However, not all key players in atherosclerotic lesion have been described among cells migrating to the atherosclerotic plaque. In my opinion, NK cells (doi: 10.3389 / fimmu.2019.01503) and NKT cells (doi: 10.1016 / j.atherosclerosis.2018.11.027.) may play an important pathogenic role in the proatherogenic process and the instability of plaques. Readers might be interested in learning about the role of these cells in atherosclerosis.
Author Response
We thank the reviewer for the positive assessment.
- We decided to introduce the diversity of T and B cell populations to illustrate to the reader that an anti-inflammatory immune response is needed to prevent cardiovascular disease by vaccination. This immune response differs from the favorable pro-inflammatory host immune responses against pathogens after vaccination.
- The reviewer is correct, more immune cells than listed play a role in the progression and modulation of atherosclerosis. We wanted to to keep the review concise and did not want to overwhelm the reader with detailed background on other leukocytes such as macrophages and myeloid cells in general. We agree that NKT cells play an important role as pro-atherosclerotic mediators and included a short paragraph including the suggested reference.
Page 3 lines 125-127:
“Natural killer T cells (NKT) - well-known drivers of atherosclerosis - have a limited TCR repertoire and are activated by endogenous and exogenous lipid antigens presented in CD1d molecules [46, 47].”
NK cells are important cytotoxic leukocytes. As the reviewer mentioned, their number are elevated in human atherosclerotic plaques and they are more inflammatory. Initially, reports demonstrated NK Cells to be pro-atherosclerotic, as depletion reduced experimental atherosclerosis. However, this effect was likely due to simultaneous targeting of NKT cells with the non-specific depletory antibodies. Recently, Nour-Erdine et al. (doi: 10.1161/CIRCRESAHA.117.311743. Epub 2017 Oct 18) convincingly showed that genetic depletion of NK cells did not affect atherosclerosis in mice. We included now a sentence mentioning the presence of NK cells in atherosclerotic aortas and plaques.
Page 2 lines 75-76:
“Hyperresponsive natural killer (NK) cells have been detected in human atherosclerotic plaques [30], albeit they do not seem to contribute to experimental atherosclerosis [31,32].“
Reviewer 3 Report
Here, authors comprehensively describe the present knowledge about vaccination strategy for the treatment of atherosclerosis. In addition, authors describe the ongoing vaccine development and approaches for atherosclerosis treatment. Importantly, authors discuss limitation and challenges needed to tackle in this area to develop clinically useful vaccine for atherosclerosis.
However, it will be useful to include briefly status of Vaccination against apolipoprotein C-III (ApoC3) in this review.
Author Response
We thank the reviewer for the positive assessment and included a paragraph about vaccination efforts against apolipoprotein C-III (ApoC3).
Page 6 lines 270-278:
“Another promising strategy is targeting Apolipoprotein CIII (APOC3), a key regulator of plasma triglycerides and independent risk factor for cardiovascular disease. APOC3 can contribute to hypertriglyceridemia by inhibiting hydrolysis of triglyceride-rich lipoproteins [98] and reducing hepatic uptake of triglyceride-rich lipoproteins [99]. Individuals with reduced circulating APOC3 levels have a reduced risk to develop coronary heart disease [100, 101] and administration of antisense DNA targeting APOC3 successfully reduced triglyceride levels [102]. Although a permanent reduction of APOC3 levels by a vaccination with virus-like particles has successfully reduced plasma triglyceride levels in mice [103], the effects of an APOC3-trageting vaccine on atherosclerosis and cardiovascular disease remain unknown.”